# Effects of β-Phenylethylamine on Psychomotor, Rewarding, and Reinforcing Behaviors and Affective State: The Role of Dopamine D1 Receptors

**DOI:** 10.3390/ijms22179485

**Published:** 2021-08-31

**Authors:** In Soo Ryu, Oc-Hee Kim, Ji Sun Kim, Sumin Sohn, Eun Sang Choe, Ri-Na Lim, Tae Wan Kim, Joung-Wook Seo, Eun Young Jang

**Affiliations:** 1Pharmacology and Drug Abuse Research Group, Korea Institute of Toxicology, Daejeon 34114, Korea; insooryu@biorchestra.com (I.S.R.); ochee.kim@kitox.re.kr (O.-H.K.); js960@naver.com (J.S.K.); rnlim@kitox.re.kr (R.-N.L.); paloma@kitox.re.kr (T.W.K.); jwseo@kitox.re.kr (J.-W.S.); 2Department of Biological Sciences, Pusan National University, Busan 46241, Korea; soomin@pusan.ac.kr (S.S.); eschoe@pusan.ac.kr (E.S.C.)

**Keywords:** β-phenylethylamine, conditioned place preference, dopamine, dopamine D1 receptor, self-administration, ultrasonic vocalization, stereotypic behavior

## Abstract

Beta-phenylethylamine (β-PEA) is a well-known and widespread endogenous neuroactive trace amine found throughout the central nervous system in humans. In this study, we demonstrated the effects of β-PEA on psychomotor, rewarding, and reinforcing behaviors and affective state using the open-field test, conditioned place preference (CPP), self-administration, and ultrasonic vocalizations (USVs) paradigms. We also investigated the role of the dopamine (DA) D1 receptor in the behavioral effects of β-PEA in rodents. Using enzyme-linked immunosorbent assay (ELISA) and Western immunoblotting, we also determined the DA concentration and the DA-related protein levels in the dorsal striatum of mice administered with acute β-PEA. The results showed that acute β-PEA increased stereotypic behaviors such as circling and head-twitching responses in mice. In the CPP experiment, β-PEA increased place preference in mice. In the self-administration test, β-PEA significantly enhanced self-administration during a 2 h session under fixed ratio (FR) schedules (FR1 and FR3) and produced a higher breakpoint during a 6 h session under progressive ratio schedules of reinforcement in rats. In addition, acute β-PEA increased 50-kHz USV calls in rats. Furthermore, acute β-PEA administration increased DA concentration and p-DAT and TH expression in the dorsal striatum of mice. Finally, pretreatment with SCH23390, a DA D1 receptor antagonist, attenuated β-PEA-induced circling behavior and β-PEA-taking behavior in rodents. Taken together, these findings suggest that β-PEA has rewarding and reinforcing effects and psychoactive properties, which induce psychomotor behaviors and a positive affective state by activating the DA D1 receptor in the dorsal striatum.

## 1. Introduction

Beta-phenylethylamine (β-PEA) is well known as an endogenous neuroactive trace amine and is widespread throughout the central nervous system of rodents [1]. In particular, β-PEA is synthesized within nigrostriatal and mesolimbic dopamine (DA) regions, such as the striatum and nucleus accumbens [2], and the highest level of β-PEA is found in these brain regions [3,4]. In addition, β-PEA is structurally similar to amphetamine. Thus, it is known that β-PEA inhibits the uptake of DA and promotes the release of DA in the mesocorticolimbic pathway [5,6]. Hossain et al. also showed that β-PEA increased extracellular DA via the DA transporter (DAT) and that increased DA levels induced by β-PEA were inhibited by a DAT inhibitor in *Caenorhabdits elegans* DAT-1 expressing LLC-pk1 cells [7]. In addition, in previous studies, systemic injection or intrastriatal infusion of β-PEA increased DA levels in the striatum [8] and nucleus accumbens [9], and β-PEA and amphetamine induced similar levels of DA release in neuronal culture [10].

The addictive behavioral effects of psychostimulants (e.g., cocaine or amphetamine) or structurally related substances [11] are associated with the activation of the mesolimbic DA system, projecting from the ventral tegmental area to the dorsal or ventral striatum. According to previous findings, β-PEA showed a reinforcing effect in self-administration studies on monkeys or dogs [12,13,14]. In addition, β-PEA produced hyperlocomotor and stereotypic behaviors in wild-type mice, and increased locomotion decreased in DAT knock-out mice [8].

Drug addiction is characterized by a drug-induced positive affective state, followed by a withdrawal-associated negative affect. Ultrasonic vocalizations (USVs), an index of the affective state, have been used in preclinical models of substance abuse [15] and in studies of motivation leading to craving or drug use in rodents [16]. Rats vocalize in distinct ultrasonic frequencies: 22-kHz USVs (low-frequency) and 50-kHz USVs (high-frequency USVs), reflecting negative and positive affective states, respectively. The 22-kHz USVs are emitted under exposure to aversive stimuli such as electrical foot shock [16,17] or drug withdrawal [18], while the 50-kHz USVs are produced by rewarding stimuli including food intake, exposure to addictive drugs, and sexual behaviors [19,20]. In previous studies, rats emitted 50-kHz USVs during amphetamine or methamphetamine self-administration [21,22] and locomotor activity was recorded after methamphetamine administration [23]. The 50-kHz USVs have also been shown to be positively correlated with extracellular DA levels in the mesolimbic system. In support of this, amphetamine increased the 50-kHz USV calls, and this effect was attenuated by a DA D1 receptor (DAD1R) antagonist in rats [24].

The dorsal striatum is a forebrain structure of the basal ganglia circuit that receives dopaminergic projections from the substantia nigra or ventral tegmental area [25]. As mentioned above, the activation of these dopaminergic pathways plays a crucial role in addictive drug-mediated motivation and reward behavior through DA-receptor-mediated neurotransmission [26,27]. In particular, DAD1R-mediated signaling is important in addictive-like behaviors [28,29]. In support of this, amphetamine-induced locomotor activity was significantly attenuated in DA D1-deficient mice [30], and the rewarding effect produced by cocaine or methamphetamine was blocked by a D1R antagonist in a conditioned place preference (CPP) study [30].

As mentioned above, others have evaluated the abuse potentials of β-PEA by measuring the behavioral and neurochemical changes in rodents; however, the mechanism underlying this effect is unknown. Therefore, in this study, we demonstrated the effects of β-PEA on addictive behaviors and emotional states and revealed the mechanisms underlying β-PEA-induced addictive-like behaviors in rodents. First, we performed an open-field test to determine whether acute systemic administration of β-PEA induces psychomotor function in mice. Second, we measured the rewarding effect of β-PEA in mice using the CPP test. Third, for the first time, we examined the effects of β-PEA on affective state by analyzing USVs. Fourth, we determined whether β-PEA produces a reinforcing effect using a rat self-administration paradigm. In addition, we measured the DA concentration and DA-related proteins in the striatum of acute β-PEA-treated mice to determine whether altered behaviors induced by β-PEA are correlated with DA. Finally, we investigated the involvement of DAD1R in β-PEA-induced addictive-like behaviors.

## 2. Results

### 2.1. Acute β-PEA Significantly Increased Circling and Head-Twitching Behaviors in Mice

To determine the psychomotor effects of β-PEA, we performed an open-field test after mice received an acute administration of saline or β-PEA. The timeline for the open-field test is illustrated in Figure 1A. The results showed that 50 mg/kg β-PEA, but not 3, 10, and 30 mg/kg, significantly increased the circling behavior compared to the saline control group (Figure 1B,C; *F*_(4,40)_ = 10.74, *p* < 0.001, 50 mg/kg β-PEA: *p* < 0.001). Similarly, 50 mg/kg β-PEA also significantly increased circling behaviors at the 10 min time point compared to the saline control group (Figure 1B,D; Time: *F*_(5,200)_ = 8.47, *p* < 0.001; Treatment: *F*_(4,40)_ = 10.74, *p* < 0.001; Interaction: *F*_(20,200)_ = 7.75, *p* < 0.001). Similar to circling behaviors, 50 mg/kg β-PEA also significantly increased head-twitching responses compared to the saline control group (Figure 1E; *F*_(4,40)_ = 26.09, *p* < 0.001, 50 mg/kg β-PEA: *p* < 0.001). Temporal analysis showed that 30 and 50 mg/kg β-PEA significantly increased head-twitching responses at the 5–15 min time point compared to the saline control group (Figure 1F; Time: *F*_(5,200)_ = 31.66, *p* < 0.001; Treatment: *F*_(4,40)_ = 25.18, *p* < 0.001; Interaction: *F*_(20,200)_ = 13.20, *p* < 0.001). Finally, β-PEA at a dose of 50 mg/kg had no effect on the locomotor activity at any time point compared to the saline control group (Figure 1G,H; *F*_(4,40)_ = 1.44, *p* = 0.24).

### 2.2. β-PEA Significantly Produced Conditioned Place Preference in Mice

We performed the CPP test to determine the rewarding effect of β-PEA. The results showed that there were no significant differences in time spent on the drug-paired side among all groups during the pre-conditioning phase (Figure 2B; *F*_(4,45)_ = 0.15, *p* = 0.96). In the post-conditioning phase, conditioning with 50 mg/kg β-PEA significantly increased the time spent on the drug-paired side compared to the saline-conditioning group (Figure 2C; *F*_(4,45)_ = 4.40, *p* < 0.01, 50 mg/kg β-PEA: *p* < 0.01). Analysis of the CPP score indicated that the mice conditioned with 50 mg/kg β-PEA had a significant increase in place preference compared to the saline-conditioned group (Figure 2D; *F*_(4,45)_ = 2.65, *p* < 0.05, 50 mg/kg β-PEA: *p* < 0.05).

### 2.3. β-PEA Significantly Increased the 50-kHz Ultrasonic Vocalization Calls in Rats

Since acute β-PEA at a dose of 50 mg/kg induced psychomotor activities such as circling or head-twitching behavior and place preference in mice, we performed the USV test to determine whether acute β-PEA at a dose of 50 mg/kg produces a positive affective state. The results showed that β-PEA significantly increased the number of 50-kHz USV calls compared to the saline control group (Figure 3B,C; *t*_(14)_ = 2.89, *p* < 0.05). A temporal analysis of USV calls revealed that acute β-PEA administration significantly increased the 50-kHz USV calls at the 10 min time point compared to that of the saline control group (Figure 3D; Time: *F*_(5,70)_ = 1.02, *p* = 0.41; Treatment: *F*_(1,14)_ = 7.11, *p* < 0.05; Interaction: *F*_(5,70)_ = 0.97, *p* = 0.44; 10 min time point: *p* < 0.05).

### 2.4. β-PEA Was Self-Administered under Fixed Ratio Schedules of Reinforcement in Rats

We used a β-PEA self-administration paradigm under FR and progressive ratio (PR) schedules of reinforcement to determine whether β-PEA produces reinforcing effects in rats. The experimental timeline for the self-administration of β-PEA under FR (FR1 and FR3) and PR schedules is illustrated in Figure 4A. The results showed that in the 1.0 mg/kg/infusion β-PEA self-administered group, there was a significantly increased number of infusions (Figure 4B; Time: *F*_(9,252)_ = 3.07, *p* < 0.01; Treatment: *F*_(3,28)_ = 29.99, *p* < 0.001; Interaction: *F*_(27,252)_ = 3.64, *p* < 0.001) and active lever pressing response (Figure 4C; Time: *F*_(9,252)_ = 5.51, *p* < 0.001; Treatment: *F*_(3,28)_ = 26.87, *p* < 0.001; Interaction: *F*_(27,252)_ = 8.35, *p* < 0.001) under the FR1 and FR3 schedules compared to the self-administered saline group. However, the number of inactive lever press responses was not altered among the groups throughout the sessions (Figure 4D). The self-administered 1.0 mg/kg/infusion β-PEA group showed a significantly increased mean number of infusions (*F*_(3,28)_ = 28.81, *p* < 0.001, 1.0 mg/kg/infusion β-PEA: *p* < 0.001; Figure 4E) and amount of drug intake (*F*_(3,28)_ = 41.22, *p* < 0.001, 1.0 mg/kg/infusion β-PEA: *p* < 0.001; Figure 4F) under the FR1 schedule compared to the self-administered saline group. Similar to the FR1 sessions, the self-administered 1.0 mg/kg/infusion β-PEA rats also showed a significantly increased mean number of infusions (Figure 4G; *F*_(3,28)_ = 24.00, *p* < 0.001, 1.0 mg/kg/infusion β-PEA: *p* < 0.001) and amount of drug intake (Figure 4H; *F*_(3,28)_ = 25.35, *p* < 0.001, 1.0 mg/kg/infusion β-PEA: *p* < 0.001) under the FR3 schedule compared to the self-administered saline group. In the temporal analysis of infusion patterns, the self-administered 1.0 mg/kg/infusion β-PEA group showed significantly increased infusions in a 20-min time interval compared to the self-administered saline group (Figure 4I,J; Time: *F*_(5,140)_ = 7.35, *p* < 0.001; Treatment: *F*_(3,28)_ = 34.36, *p* < 0.001; Interaction: *F*_(15,140)_ = 1.44, *p* = 0.13).

### 2.5. β-PEA Produced a Higher Breakpoint in Self-Administration under Progressive Ratio Schedules of Reinforcement in Rats

After 10 sessions of FR schedules of reinforcement, rats self-administered β-PEA under a PR schedule of reinforcement to determine the reinforcing efficacy of β-PEA. The results showed that there was a significant increase in the total number of infusions in the self-administered 1.0 mg/kg/infusion β-PEA group (Figure 5A; *F*_(3,28)_ = 38.57, *p* < 0.001, 1.0 mg/kg/infusion β-PEA: *p* < 0.001) under a PR session compared to the self-administered saline group. Similarly, the breakpoint of the self-administered 1.0 mg/kg/infusion β-PEA group under a PR session was also significantly increased compared to that of the self-administered saline group (Figure 5B; *F*_(3,28)_ = 52.63, *p* < 0.001, 1.0 mg/kg/infusion β-PEA: *p* < 0.001). The analysis of lever responses under a PR session showed that the active lever pressing responses in the self-administered 1.0 mg/kg/infusion β-PEA group were also significantly increased compared to those of the self-administered saline group (Figure 5C,D; *F*_(3,28)_ = 51.92, *p* < 0.001, 1.0 mg/kg/infusion β-PEA: *p* < 0.001); however, inactive lever pressing responses were not altered among the groups. In the temporal analysis of infusion patterns, the self-administered 1.0 mg/kg/infusion β-PEA group significantly increased infusions at the 1 h time point compared to the self-administered saline group (Figure 5E,F; Time: *F*_(5,140)_ = 88.06, *p* < 0.001; Treatment: *F*_(3,28)_ = 50.18, *p* < 0.001; Interaction: *F*_(15,140)_ = 30.50, *p* < 0.001; 1 h time point: *p* < 0.001).

### 2.6. Acute β-PEA Administration Significantly Increased DA Concentration and p-DAT and TH Expression in the Dorsal Striatum of Mice

Since acute β-PEA administration increased psychoactive behaviors and produced a positive affective state, we investigated the DA concentration and expression of DA-related proteins in the dorsal striatum of β-PEA-administered mice to determine whether β-PEA alters dopaminergic neurotransmission in the dorsal striatum (Figure 6A). The results showed that acute β-PEA significantly increased the DA concentration in the dorsal striatum compared to the saline control group (Figure 6B; *t*_(8)_ = 3.01, *p* < 0.05). Additionally, acute β-PEA administration significantly increased the immunoreactivities of phospho (p)-DAT (*t*_(8)_ = 2.59, *p* < 0.05) and tyrosine hydroxylase (TH) (*t*_(8)_ = 2.41, *p* < 0.05) in the dorsal striatum compared to the saline control group (Figure 6C,D,F). However, there was no difference in the immunoreactivities of DAT and vesicular monoamine transporter 2 (VMAT-2) levels in the dorsal striatum of β-PEA-administered mice compared to the saline control group (Figure 6C,E,G).

### 2.7. Blockade of DAD1R Attenuated Acute β-PEA-Induced Circling Behavior in Mice

Since acute β-PEA administration increased the DA concentration in the dorsal striatum, we investigated whether the pharmacological blockade of DAD1R alters β-PEA-induced increases in circling and head-twitching behaviors using an open-field test. The experimental timeline for the open-field test following systemic pretreatment with SCH23390 followed by acute β-PEA administration is illustrated in Figure 7A. As shown in Figure 7B,C, the pretreatment with SCH23390 (0.02 mg/kg) significantly attenuated circling behavior (Figure 7B; *t*_(8)_ = 2.59, *p* < 0.05 and Figure 7C; Time: *F*_(5,50)_ = 33.23, *p* < 0.001; Treatment: *F*_(1,10)_ = 5.33, *p* < 0.05; Interaction: *F*_(5,50)_ = 3.95, *p* < 0.01; 10 min time point: *p* < 0.001) but not a head-twitching response (Figure 7D,E) or locomotor activity (Figure 7F,G) in β-PEA-administered mice. In addition, the pretreatment with SCH23390 alone did not alter the distance traveled compared to the vehicle pretreatment group (Appendix A), indicating that SCH23390 did not affect general behavior.

### 2.8. Blockade of DAD1R Significantly Decreased β-PEA-Taking Behavior

Since the stimulation of DAD1R is closely associated with the enhancement of dopaminergic neurotransmission in the brain, leading to the reinforcing effects of drugs [31,32], we investigated whether the pharmacological prevention of DAD1R alters the reinforcing effects of β-PEA using a self-administration paradigm. The experimental timeline is illustrated in Figure 8A. As shown in Figure 8B,C, rats were trained to press the active lever associated with drug infusion and maintained stable responses of active lever pressing and infusion of β-PEA without any difference in the inactive lever response in a time-dependent manner. In the last three days of the training period, the number of infusions and active lever responses were consecutively maintained (Figure 8E). In the final session of self-administration, the results showed that pretreatment with SCH23390 significantly decreased the number of infusions (Figure 8F; *t*_(10)_ = 6.15, *p* < 0.001) and active lever responses (Figure 8G; *t*_(10)_ = 6.72, *p* < 0.001) compared to the vehicle pretreatment group. Similarly, in the temporal analysis, pretreatment with SCH23390 significantly decreased the number of infusions at 60, 80, 100, and 120 min (Figure 8H,I; Time: *F*_(5,60)_ = 2.75, *p* < 0.05; Treatment: *F*_(1,12)_ = 16.06, *p* < 0.01; Interaction: *F*_(5,60)_ = 6.85, *p* < 0.001) after the beginning of the self-administration session.

## 3. Discussion

In the present study, we demonstrated the psychoactive properties of β-PEA and their underlying mechanisms using multiple behavioral and emotional state assessments in rodents. The results showed that acute β-PEA administration significantly increased circling and head-twitching behaviors and produced drug-paired place preference in mice. In addition, the exposure to β-PEA induced a positive affective state in rats, and rats self-administered β-PEA under FR and PR schedules of reinforcement. In mechanistic studies, the administration of β-PEA significantly increased the DA concentration and TH and p-DAT protein expression in the dorsal striatum of mice. Finally, the pharmacological blockade of DAD1R with a DAD1R antagonist, SCH23390, significantly attenuated circling behavior in mice and β-PEA-taking behavior in rats. Taken together, our findings suggest that β-PEA has psychoactive properties given its rewarding and reinforcing effects via the activation of dopaminergic neurotransmission, and it has psychotomimetic potentials such as circling or head-twitching behavior. These effects are likely mediated by DAD1R in the dorsal striatum of rodents.

In general, the increase in dopaminergic neurotransmission in the striatum is closely associated with drug-induced increases in psychoactive behaviors such as hyperactivity, place preference, and drug-taking behavior [33,34]. Amphetamine and amphetamine-like drugs induce DA efflux through a DAT channel. DAT can also control the dynamics of dopaminergic neurotransmission by driving reuptake of extracellular DA into presynaptic neurons and by phosphorylation at Thr53 residue to reverse its function [35,36]. In the present study, acute β-PEA administration increased striatal DA concentration and TH, a rate-limiting enzyme in DA synthesis, and p-DAT expression in mice, which are well known as modulators of DA concentration in the reward system. Based on our results, it is predicted that acute β-PEA administration induces p-DAT, reverses the function of DAT, and finally increases DA neurotransmission in the striatum. However, unexpectedly, we found that despite the increase in striatal DA concentration by acute administration of β-PEA (50 mg/kg), there was no change in locomotor activity. This result is partially inconsistent with another study [8]. Sontnikova et al. showed that β-PEA increased locomotor activity and stereotypical behaviors such as grooming with an increase in striatal DA levels in mice [8]. On the other hand, in the present study, β-PEA at a high dose of 50 mg/kg increased circling, known as stereotyped behavior, and the head-twitching response, considered a behavioral marker of the hallucinogen effect in humans via hyperstimulation of dopaminergic neurotransmission. Previous studies reported that β-PEA derivatives such as N-2-methoxybenzyl-phenethylamines (NBOMes) also produced head-twitching response. For example, 25B-NBOMe (4-bromo-2,5-dimethoxy-N-(2-methoxybenzyl)phenethylamine) decreased locomotor activity in the open field test and induced hallucinogenic activity, such as head and body twitch responses [37]. In addition, 25I-NBOMe (4-iodo-2,5-dimethoxy-N-(2-methoxybenzyl)phenethylamine) also displayed head shaking [38] and decreased locomotor activity [39]. Considering the head-twitching response, which is a rapid side-to-side rotational head movement without changing its position (nonmoving state), it could be thought that no change in locomotor activity was due to a lack of time to move by excessive head-twitching responses. Furthermore, it is known that behaviors such as circling or head-twitching response are associated with the mesocortical dopaminergic system [40,41]. Thus, we need to investigate the effect of behavior produced by β-PEA and its underlying mechanisms in a future study.

The CPP test is a classic procedure used in animals to evaluate the rewarding effect of contextual stimuli associated with exposure to addictive drugs [42,43], and intravenous (i.v.) self-administration in rodents is a useful model for predicting the abuse liability of novel drugs in humans [43]. Only one study demonstrated that β-PEA induced a CPP in dogs [4], and a few studies previously reported that PEA derivatives and NBOMes produced rewarding and reinforcing effects via stimulation of dopaminergic neurotransmission in CPP and self-administration studies [44,45]. Consistent with a previous finding, our results demonstrated that 50 mg/kg β-PEA significantly increased DA concentration in the dorsal striatum and induced a drug-paired place preference in a dose-dependent manner. In our self-administration study, a high dose of β-PEA (1.0 mg/kg/infusion) was reliably self-administered under FR reinforcement schedules. We demonstrated the reinforcing effect of β-PEA in a rat self-administration study for the first time in the present study. This result is consistent with a previous study showing the psychomotor stimulant effects of β-PEA in monkeys [14]. In monkey self-administration study, inhibition of monoamine oxidase-type B enhanced the discriminative or reinforcing-stimulus effects of β-PEA, presumably by delaying its inactivation. In addition, the PR schedule of reinforcement in self-administration paradigms has been used to directly measure the strength of reinforcement (i.e., how hard the animal will work) of psychostimulants [46]. In the present study, we tested the strength of reinforcement of β-PEA for the first time. Our results demonstrate that the self-administration of 1.0 mg/kg/infusion β-PEA under the PR schedule following the FR schedules significantly increased drug-taking behavior, and the breakpoint of 1.0 mg/kg/infusion β-PEA was similar to that of other psychoactive drugs [47,48]. Taken together, our results indicate that β-PEA can act as a rewarding stimulus and a positive reinforcer via activation of the striatal dopaminergic system.

USVs recording is a useful method to measure changes in emotional state caused by exposure to drugs such as cocaine or methamphetamine [49,50]. It is well known that the increase in frequency of 50-kHz USV calls indicates a positive mood state, which is also closely related to stimulation of dopaminergic neurotransmission in the reward system [51]. For example, amphetamine increased 50-kHz USVs, which were inhibited by the blockade of the DAD1R antagonist SCH23390 in rats [52]. Consistently, our results demonstrated that acute β-PEA administration at a dose of 50 mg/kg significantly increased the frequency of 50-kHz USV calls. Based on these findings, one could speculate that β-PEA also produces a positive affective state via stimulation of the striatal dopaminergic system.

DAD1R is widely distributed in the brain, including the dorsal striatum, and plays a critical role in behavioral and emotional alterations in response to hyperactivation of dopaminergic neurotransmission elicited by exposure to psychostimulants [52,53,54]. Previous studies reported that the pharmacological blockade of DAD1R attenuated cocaine-induced locomotion, including locomotor activity [55,56] and the rewarding effect induced by d-amphetamine and cocaine [57]. In addition, DAD1R knockout mice did not self-administer cocaine [58], and SCH23390, a DAD1R antagonist, attenuated cocaine-, methamphetamine- or fentanyl-taking behavior in self-administration of rats [59,60,61]. Consistently, the results of the present study demonstrate that pretreatment with SCH23390 significantly attenuated circling behavior induced by acute β-PEA and β-PEA-taking behavior under the FR schedule in the self-administration tests. This indicates that DAD1R-mediated signaling pathways may contribute to the expression of β-PEA-induced addictive behaviors. However, in this study, the blockade of DAD1R decreased β-PEA-induced circling behavior but not the head-twitching response. In particular, the head-twitching response is also associated with the activation of the serotonergic system in the frontal cortex, and the activation of the 5-HT_2A_ receptor induces head-twitching response in rodents [38]. Thus, as mentioned above, further studies are needed to investigate the hallucinogen effect of β-PEA and its underlying mechanism.

In summary, we demonstrated that β-PEA has rewarding and reinforcing effects and induces stereotypical behaviors and a positive affective state by increasing the DA concentration and expression of DA-related proteins (TH and p-DAT) in the striatum of rodents. In addition, circling behavior and β-PEA-taking behavior are attenuated by blockade of DAD1R. Therefore, our results suggest that the activation of DAD1R is important for β-PEA-induced addictive behaviors.

## 4. Materials and Methods

### 4.1. Animals

Male C57BL/6 mice (18–25 g) used for open-field, CPP tests, DA enzyme-linked immunosorbent assay (ELISA), Western blotting, and male Sprague-Dawley rats (230–270 g) used for self-administration and USV tests were purchased from Orient Bio. Inc. (Seongnam, Korea). Animals were housed in a controlled environment (temperature: 23–25 °C, humidity: 30–70%) under a regular 12-h light/dark cycle (lights on at 8:00 a.m.). Mice were housed in a group of 4 animals, and rats were housed individually for the self-administration test or in pairs for the USV test. Animals were provided ad libitum access to food and water, except during a food training period of self-administration procedures. During the food training period, rats were food-restricted to 13–15 g/day to increase the probability of lever responding [62]. All animal experiments were conducted in a quiet room to minimize environmental stress during the light cycle. All experimental procedures were approved by the Institutional Animal Care and Use Committee of Korea Institute of Toxicology and were conducted in accordance with the provisions of the National Institutes of Health Guide for the Care and Use of Laboratory Animals.

### 4.2. Drugs

β-PEA and a DAD1R antagonist, SCH23390, were purchased from Sigma-Aldrich (St. Louis, MO, USA). β-PEA and SCH23390 were dissolved in 0.9% physiological saline. The intraperitoneal (i.p.) injection for SCH23390 was determined based on a previous study [63]. For self-administration studies, a working solution of saline or β-PEA was filtered through a syringe-mounted 0.22-µm Minisart^®^ Syringe Filter (Sartorius Stedim Biotech, Goettingen, Germany) immediately before use. All drug solutions were prepared immediately prior to the beginning of each experiment.

### 4.3. Open Field Test

The open field test was performed as previously described [48]. The timeline for the open-field test after acute saline or β-PEA is illustrated in Figure 1A. Mice were acclimated to an open-field arena in illuminated and sound-attenuated cubicles for at least 6 days (for 30 min/day) to minimize environmental influences. On the test day, mice were placed in the open-field arena, and then basal activity was measured for 30 min. After recording basal activity, the mice received acute i.p. injection of saline (10 mL/kg) or β-PEA (3, 10, 30, or 50 mg/kg, 10 mL/kg) (Figure 1A). Behavioral activities were additionally recorded for 30 min after the acute administration of saline or β-PEA. Circling behavior (active movement in a circular direction), head-twitching response (a rapid left to right or right to left movement of the head), and locomotor activity (total distance traveled) were recorded using a computer-based monochrome/near infrared video camera (Med Associates, Georgia, VT, USA). Changes in circling behavior (count) and locomotor activity (centimeter) were quantified using a computer-based video tracking system (Ethovision XT14, Noldus, Wageningen, Netherlands). Chances in head-twitching response (count) were quantified by monitoring two trained observers (blinded to the treated groups).

To determine the involvement of DAD1R in β-PEA-induced behavioral alterations, another cohort of mice was pretreated with vehicle or SCH23390 (0.02 mg/kg, 10 mL/kg, i.p.) 30 min before acute administration of saline or β-PEA (50 mg/kg, i.p.) (Figure 7A), and then mice were observed for behavioral activities over 30 min.

### 4.4. Conditioned Place Preference Test

The CPP paradigm was performed as described in our previous study [48,62]. The CPP apparatus (MED-CPP-3013–2, Med Associates) consisted of two large compartments (17.4 cm × 12.7 cm × 12.7 cm) separated by a guillotine door. One compartment was a black room with a stainless-steel grid floor consisting of rods (diameter: 3.2 mm) placed 7.9 mm apart. The other compartment was a white room with a 6.35 mm stainless steel mesh floor. The guillotine door that provided access was placed in the center of the two conditioning compartments. Each compartment had a Plexiglas top with controlled illumination (15–20 lux). The CPP paradigm was performed according to an unbiased and counterbalanced subject assignment procedure. The timeline for the CPP experiment is illustrated in Figure 2A. Briefly, the CPP test was composed of four different phases: (1) habituation phase (Day 1 and 2, 20 min); (2) pre-conditioning phase (Day 3, 20 min); (3) conditioning phase (Day 4–11, 30 min); and (4) post-conditioning phase (Day 12, 20 min) (Figure 2A). The pre-conditioning data were used to classify the mice into groups that showed approximately equal preference for either side of the apparatus. The mice that stayed over 960 s in either compartment during the pre-conditioning phase were excluded. The cutoff value (>960 s) for biased preference was determined from our previous study [62].During the conditioning phase, the mice were given an injection of saline (10 mL/kg, i.p.) or β-PEA (3, 10, 30, or 50 mg/kg, i.p.) on alternating days (β-PEA: Days 4, 6, 8, and 10; saline: Days 5, 7, 9, and 11). Each mouse was then confined to one of the compartments after saline injection or to the other compartments after β-PEA injection for 30 min (counterbalanced). In the post-conditioning phase, each mouse was allowed to roam freely between both sides of the apparatus for 20 min. A CPP score was determined by calculating the difference between the time spent in the drug-paired side of the apparatus during the pre- versus post-conditioning phase, as described previously [48,62].

### 4.5. Recordings of Ultrasonic Vocalizations 

USVs were recorded in customized sound-attenuating chambers that consisted of two boxes to minimize exterior noise (inside box: 57 × 44.5 × 40 cm, outside box: 75 × 63 × 60 cm), as described previously [50]. Ultrasound microphones (Ultramic250K, Dodotronic, Italy) were mounted ~2.5 cm above the center of the inner chamber ceiling. The microphone signals were recorded, and wav files were analyzed using Avisoft-RECORDER software (Avisoft Bioacoustics, Berlin, Germany). The recorded audio had a sample rate of 200 kHz and a 16-bit format. Skilled researchers analyzed (blinded to the treatment groups) 50-kHz USVs through spectrographic analysis and playback (slowed down to 2%) using Avisoft-SASLab Pro (software version 5.2.13, Avisoft Bioacoustics). The USVs were band-filtered between 32 and 96 kHz, and the spectrograms were generated with a fast Fourier transform length of 512, with an overlap of 75% (FlatTop window, 100% frame size), a frequency resolution of 391 Hz, and a time resolution of 1.28 milliseconds (ms). The timeline for USVs test is illustrated in Figure 3A. In brief, rats were habituated to the test chamber for 30 min for at least 6 days to minimize the stress response to a novel environment. On the test day, rats adapted for 30 min in the customized chamber, and the USVs were then recorded for 30 min as baseline. After recording basal USVs, rats were given acute saline or 50 mg/kg β-PEA, and then the USVs were recorded for an additional 30 min. Data are expressed as the total number of 50-kHz USVs for 30 min and a 5 min time interval. The dose of β-PEA was chosen for the USV test based on the results of the psychomotor and CPP tests.

### 4.6. Food Training and Catheter Implantation Surgery for Self-Administration

Food training and catheter implantation surgery were performed as described previously [48]. The timeline for self-administration procedures is illustrated in Figure 4A. In brief, rats were trained to lever press for 45 mg food pellets (BioServ, Frenchtown, NJ, USA) under a continuous FR1 schedule of reinforcement during 1 h sessions. After the acquisition criterion (obtained more than 80 food pellets/1 h, 3 consecutive days), rats were anesthetized under 2–3% isoflurane, and then a chronic indwelling jugular catheter was inserted into the right jugular vein and secured to muscle around the jugular vein with Mersilene surgical mesh (Ethicon Inc., Somerville, NJ, USA). The distal end of the catheter was threaded subcutaneously and connected to a 22-gauge stainless steel cannula (P1 Technologies, Roanoke, VA, USA), fixed to the head assembly with dental cement and secured with Prolene surgical mesh (Ethicon Inc. Somerville, NJ, USA). After catheter implantation surgery, rats were allowed to recover from surgery for 7 days prior to the self-administration test. During recovery, the catheter was flushed once daily with 0.2 mL of heparinized saline (30 IU/mL) including gentamicin sulfate (0.33 mg/mL) to prevent clotting and infection.

### 4.7. Drug Self-Administration

The self-administration test was performed as described previously [48,62]. After recovery from i.v. catheter surgery, rats began the β-PEA self-administration test for 11 consecutive days under FR schedules of reinforcement (FR1: Day 1–7, FR3: Day 8–10, daily 2 h session) and then a PR schedule of reinforcement (Day 11, 6 h session) (Figure 4A). During self-administration sessions, a response on the active lever (drug-paired lever) resulted in a 0.1 mL i.v. delivery of saline or β-PEA (0.1, 0.3, or 1.0 mg/kg/infusion) for 4.1 s. Each infusion was followed by an additional 20 s time-out (TO) period. During the TO period, active lever responses were recorded but did not result in infusion of drug. Inactive lever presses were also recorded but had no consequence.

After 10 consecutive FR sessions, rats underwent a 6 h self-administration session of the PR schedule to assess the reinforcing efficacy of β-PEA (Figure 4A). During the PR schedule, the number of lever presses required to obtain a single infusion of β-PEA was determined by the following equation: responses per drug delivery = [5e^(injection number × 0.2)^] − 5 (i.e., 1, 2, 4, 6, 9, 12, 15, 20, 25, 32, 40, 50, 62, 77, and 95) [46]. The total lever presses and infusions were recorded throughout the session.

To determine the involvement of DAD1R in the drug-taking behavior of β-PEA self-administration, we tested the effect of SCH23390, an antagonist of DAD1R, on β-PEA-taking behavior in a separate group. The experimental timeline is represented in Figure 8A. Rats were self-administered 1.0 mg/kg/infusion β-PEA for at least 14 consecutive sessions under the FR1 schedule. When rats met a criterion (with a <10% change in the number of infusions for 3 consecutive days), they were randomly divided into two different groups as follows: (1) pretreatment with vehicle (1 mL/kg, i.p.) + β-PEA self-administration and (2) pretreatment with SCH23390 (0.02 mg/kg, i.p.) + β-PEA self-administration. On the test day, rats were pretreated with vehicle or SCH23390 followed by β-PEA self-administration under the FR1 schedule. The number of infusions and lever responses were recorded for 2 h.

### 4.8. Tissue Collection for DA ELISA and Western Blotting

Mice were given acute saline or β-PEA (50 mg/kg, i.p.) administration. Mice were anesthetized under avertin (200 mg/kg, i.p.) at 10 min after administration of saline or β-PEA (50 mg/kg, i.p.) based on the results of locomotor and USVs, and brains were rapidly removed. The brain was serially cut using a stainless-steel coronal brain matrix (Roboz Surgical Instrument Co., Inc., Gaithersburg, MD, USA) on ice, and the dorsal striatum was collected bilaterally. The tissues were transferred to a mixture of RIPA buffer (Thermo Fisher Scientific, Waltham, MA, USA) and protease and phosphatase inhibitor cocktails (Thermo Fisher Scientific, Waltham, MA, USA), homogenized and incubated on ice for 1 h. After incubation, the lysates were centrifuged at 13,000 rpm for 30 min at 4 °C. The pellet was discarded, and the supernatant was centrifuged again at 13,000 rpm for 30 min at 4 °C. For Western blotting, the concentration of the solubilized proteins in the supernatant fraction was determined based on the bicinchoninic acid (BCA) assay using a BCA assay kit (Thermo Fisher Scientific, Waltham, MA, USA). The solubilized protein samples were stored in a deep freezer before use.

### 4.9. Dopamine ELISA

The changes in DA concentration of the dorsal striatum after the acute administration of saline or β-PEA were measured using a DA ELISA kit (#KA1887, Abnova, Taipei, Taiwan). According to the manual, standards, controls, and samples (protein lysates of the dorsal striatum) were subjected to extraction and acylation assays. The acylated standards, controls, and samples were pipetted into the appropriate wells of the DA microtiter strips and incubated with DA antiserum for 30 min at room temperature on a shaker. Thereafter, the contents were discarded, and the wells were washed. The conjugate was added to all wells and incubated for 15 min, and the wells were washed. The substrate was pipetted into all wells and incubated for 15 min. After 3 washes for 10 min, the stop solution was added into the wells, and then the absorbance was measured at 450 nm in a GloMax microplate multimode reader (Promega, Madison, WI, USA). All measurements were performed in duplicate.

### 4.10. Western Immunoblotting

Western immunoblotting was performed as previously described [62]. The solubilized proteins were resolved using 10% acrylamide gel electrophoresis, after which the separated proteins were transferred to a polyvinylidene fluoride (PVDF) membrane (Bio-Rad, Hercules, CA, USA). The PVDF membrane was blocked with blocking buffer containing 5% bovine serum albumin in a mixture of Tris-buffered saline and 0.1% Tween-20 (TBST) and washed three times for 10 min each with TBST. After washing, the membrane was probed with a rabbit or a mouse primary antiserum for p-DAT (1:1000, #ab183486, Abcam, Cambridge, UK), TH (1:1000, #2792S, Cell Signaling Technology, Danvers, MA, USA), and VMAT-2 (1:1000, #sc-374079, Santa Cruz Biotechnology, Dallas, TX, USA) for 16 h at 4 °C on a shaker. Then, the membrane was rewashed three times and incubated with appropriate horseradish peroxidase-labeled secondary antiserum (Thermo Fisher Scientific, Waltham, MA, USA) for 1 h at room temperature. Immunoreactive protein bands were detected by enhanced chemiluminescence reagents (Thermo Fisher Scientific, Waltham, MA, USA) using Image Lab software (version 5.2.1, Bio-Rad). The membrane was reprobed using a rabbit primary antiserum against total DAT (1:1000, #ab111468, Abcam) after stripping the same membrane that was confirmed to contain p-DAT and reprobed for β-actin (1:2000; #A5316, Sigma-Aldrich, St. Louis, MO, USA) after stripping for blot normalization. Immunoreactive protein bands on the membrane were semiquantified using ImageJ software (version 1.52a, National Institutes of Health, Bethesda, MD, USA).

### 4.11. Statistical Analyses

Statistical analyses were performed using GraphPad Prism 8 (version 8.0.2, GraphPad Software, La Jolla, CA, USA). Data are presented as the mean ± standard error of the mean (SEM) of all experiments. Statistical significance between the groups was determined by a two-tailed unpaired *t*-test (Figure 3, Figure 6, Figure 7 and Figure 8). Tukey’s post hoc test was used for all one-way analyses of variance (ANOVAs) with repeated measures, and Bonferroni’s post hoc test was used for all repeated measures two-way ANOVAs (Figure 1, Figure 2, Figure 4 and Figure 5). The level of statistical significance was set at *p* < 0.05. Asterisks *, **, *** and #, ##, ### in the figures represent significance levels of *p* < 0.05, *p* < 0.01, and *p* < 0.001, respectively.

## Figures and Tables

**Figure 1 ijms-22-09485-f001:**
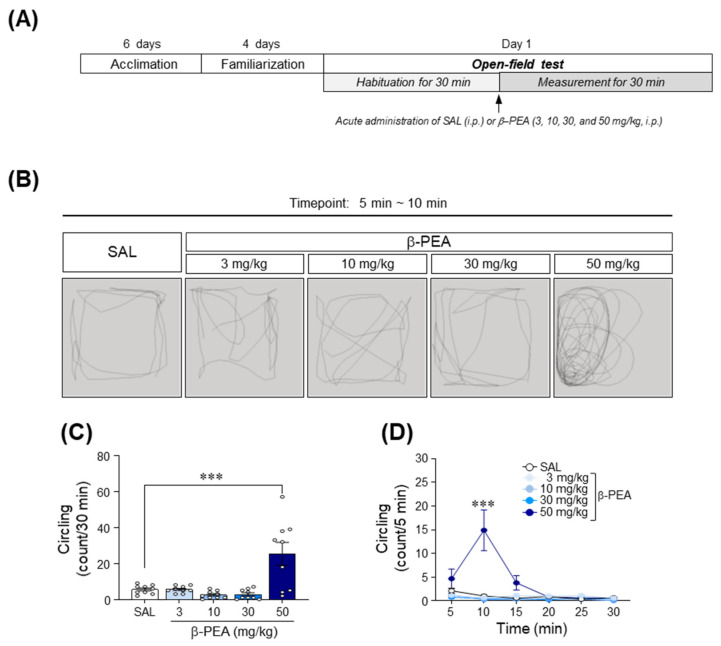
Acute β-PEA administration significantly increased circling and head-twitching behavior in mice. (**A**) Experimental timeline for open-field test. (**B**) Representative circling patterns (black lines in each gray rectangle) 5–10 min after acute administration of saline or β-PEA. (**C**,**E**,**G**) Total number of circling (**C**), head-twitching response (**E**), and distance traveled (**G**) 30 min after acute saline or β-PEA administration. (**D**,**F**,**H**) Temporal changes in circling (**D**), head-twitching response (**F**), and distance traveled (**H**) in a 5-min time interval after acute saline or β-PEA administration. * *p* < 0.05, ** *p* < 0.01, *** *p* < 0.001 vs. saline control group. SAL, saline; β-PEA, β-phenylethylamine. *n* = 9 per group.

**Figure 2 ijms-22-09485-f002:**
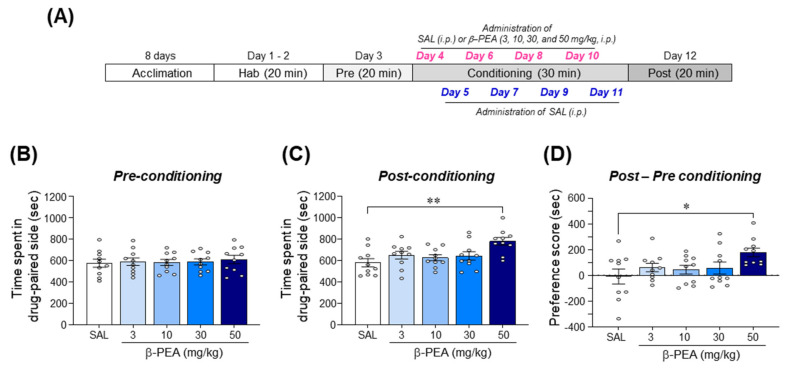
β-PEA administration significantly induced conditioned place preference (CPP) in mice. (**A**) Experimental timeline for CPP test of β-PEA. (**B**,**C**) Time spent on the drug-paired side during the pre- (**B**) or post- (**C**) conditioning phase. (**D**) CPP score (post–pre) for β-PEA. * *p* < 0.05, ** *p* < 0.01 vs. saline-conditioned group. SAL, saline; β-PEA, β-phenylethylamine. *n* = 10 per group.

**Figure 3 ijms-22-09485-f003:**
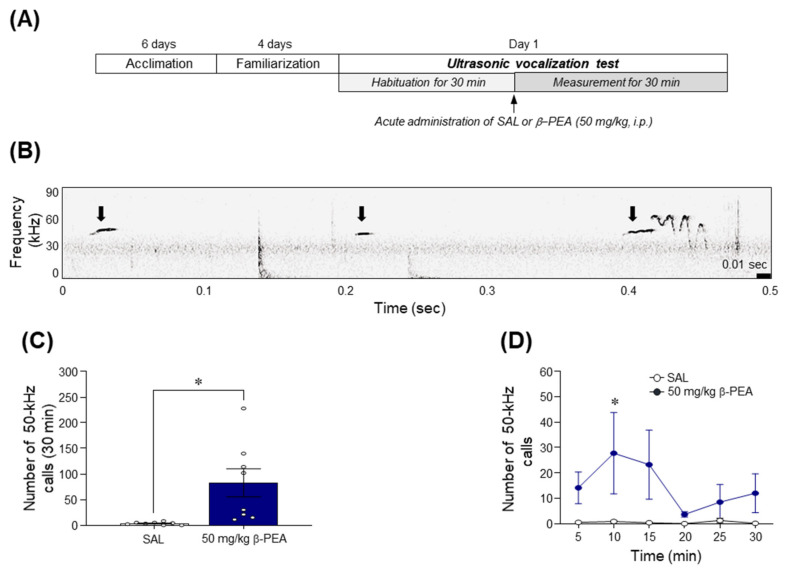
Acute β-PEA administration significantly induced 50-kHz ultrasonic vocalizations (USVs) calls in rats. (**A**) Experimental timeline for USV test. (**B**) Representative spectrograms of 50-kHz USVs after acute administration of β-PEA. Black arrows indicate exemplary spectrograms of 50-kHz USVs in rats. (**C**) Total number of calls of 50-kHz USVs for 30 min after acute administration of β-PEA. (**D**) Temporal changes in calls of 50-kHz USVs in a 5-min time interval after acute β-PEA administration. * *p* < 0.05 vs. saline control group. SAL, saline; β-PEA, β-phenylethylamine. *n* = 8 per group.

**Figure 4 ijms-22-09485-f004:**
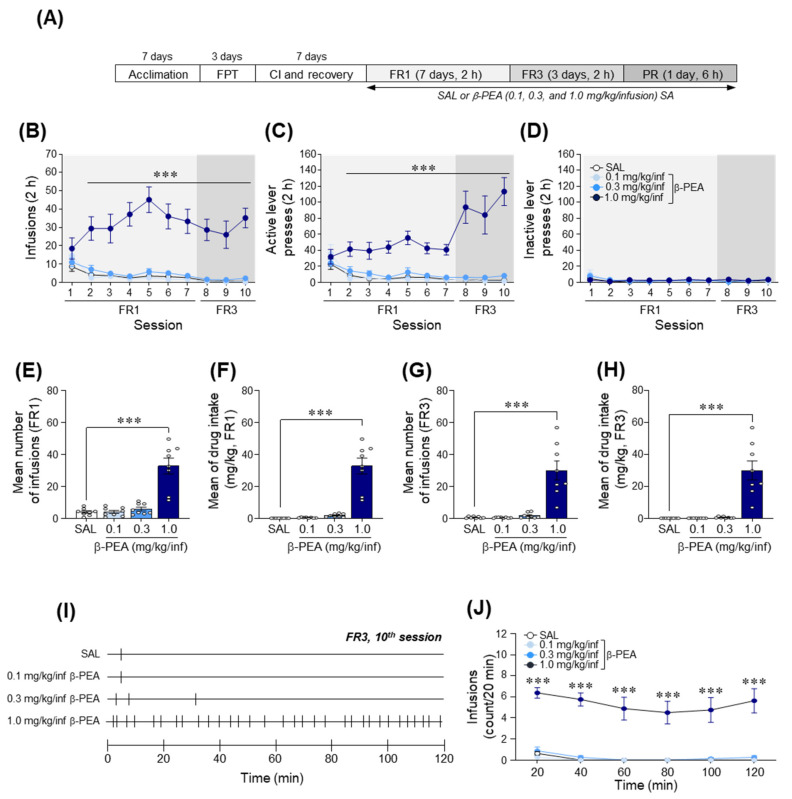
Effects of β-PEA on self-administration under fixed ratio (FR) schedules of reinforcement in rats. (**A**) Experimental timeline for β-PEA self-administration. (**B**–**D**) Number of infusions (**B**), active lever (**C**), and inactive lever (**D**) pressing responses during 2 h session β-PEA self-administration under FR1 and FR3 schedules. (**E**–**H**) Mean number of infusions and drug intake under FR1 (**E**,**F**) and FR3 (**G**,**H**) schedules of reinforcement. (**I**) Representative hatch marks indicate the infusion patterns of saline and β-PEA for 2 h at the final self-administration session of the FR3 schedule. (**J**) Temporal changes in drug infusions at the final self-administration session of the FR3 schedule. *** *p* < 0.001 vs. saline control group. SAL, saline; β-PEA, β-phenylethylamine, FPT: food training test, CI; catheter implantation. *n* = 8 per group.

**Figure 5 ijms-22-09485-f005:**
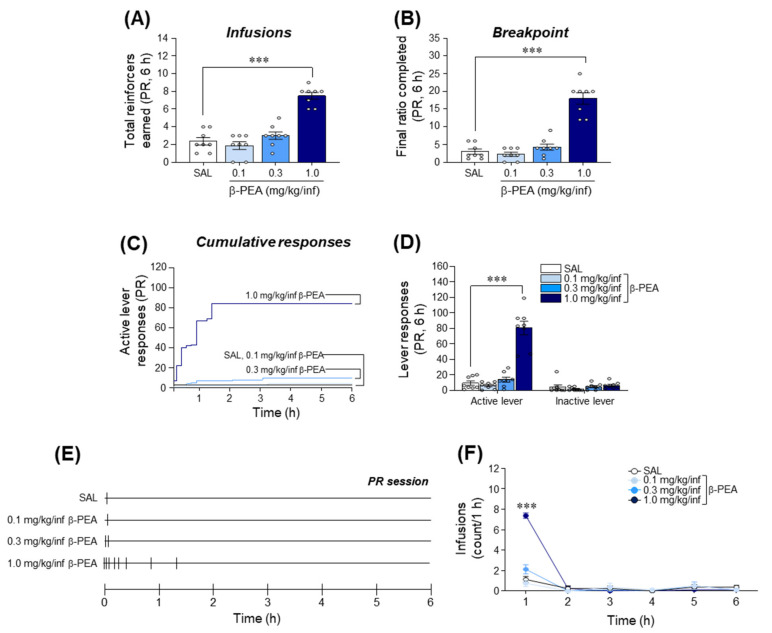
The reinforcing strength of β-PEA self-administration under the progressive ratio (PR) schedule of reinforcement in rats. (**A**) Total infusions of saline and β-PEA self-administration under the PR schedule of reinforcement. (**B**) Breakpoint of saline and β-PEA self-administration under the PR session. (**C**) Representative plots for cumulative active lever responses in the β-PEA self-administered rats during the PR session. (**D**) Number of active and inactive lever pressing responses of saline or β-PEA self-administration under the PR schedule. (**E**,**F**) Representative hatch marks (**E**) and temporal changes (**F**) in infusions at the 1.0 mg/kg/infusion β-PEA self-administered rats under the PR session. *** *p* < 0.001 vs. saline control group. SAL, saline; β-PEA, β-phenylethylamine. *n* = 8 per group.

**Figure 6 ijms-22-09485-f006:**
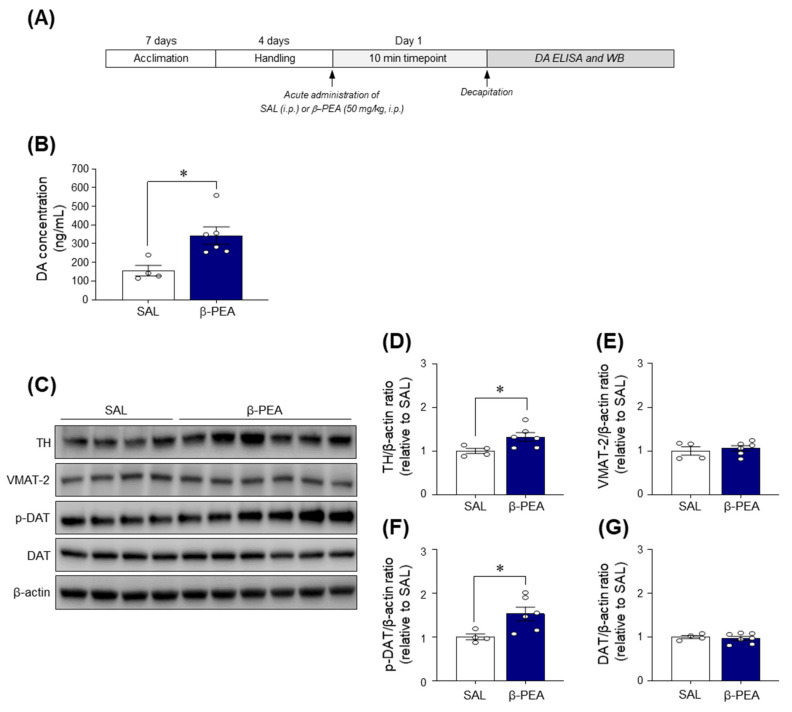
Acute β-PEA administration significantly increased DA concentrations and DA-related proteins in the dorsal striatum of mice. (**A**) Experimental timeline. (**B**) DA concentrations in the dorsal striatum of acute saline- or β-PEA-treated mice. (**C**–**G**) Immunoreactivities of TH, VMAT-2, DAT, and p-DAT proteins in the dorsal striatum of acute saline- or β-PEA-treated mice. * *p* < 0.05 vs. saline control group. SAL, saline; β-PEA, β-phenylethylamine. *n* = 4–6 per group.

**Figure 7 ijms-22-09485-f007:**
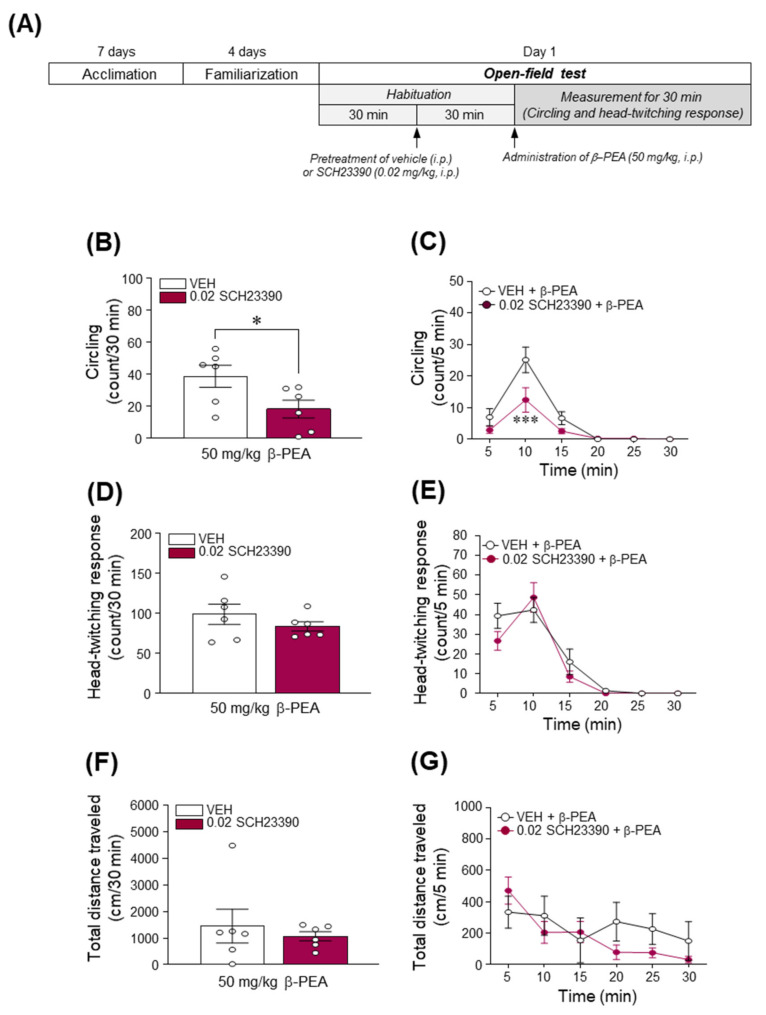
Blockade of the DA D1 receptor partially attenuated acute β-PEA-induced circling behaviors in mice. (**A**) Experimental timeline for the open-field test. (**B**–**G**) Effect of SCH23390 (0.02 mg/kg) on circling (**B**), head-twitching response (**D**), and locomotor activity (**F**) for 30 min and temporal changes in circling (**C**), head-twitching response (**E**) behaviors and locomotor activity (**G**) in a 5-min time interval after β-PEA administration. * *p* < 0.05, *** *p* < 0.001 vs. saline control group. VEH, vehicle; β-PEA, β-phenylethylamine. *n* = 6 per group.

**Figure 8 ijms-22-09485-f008:**
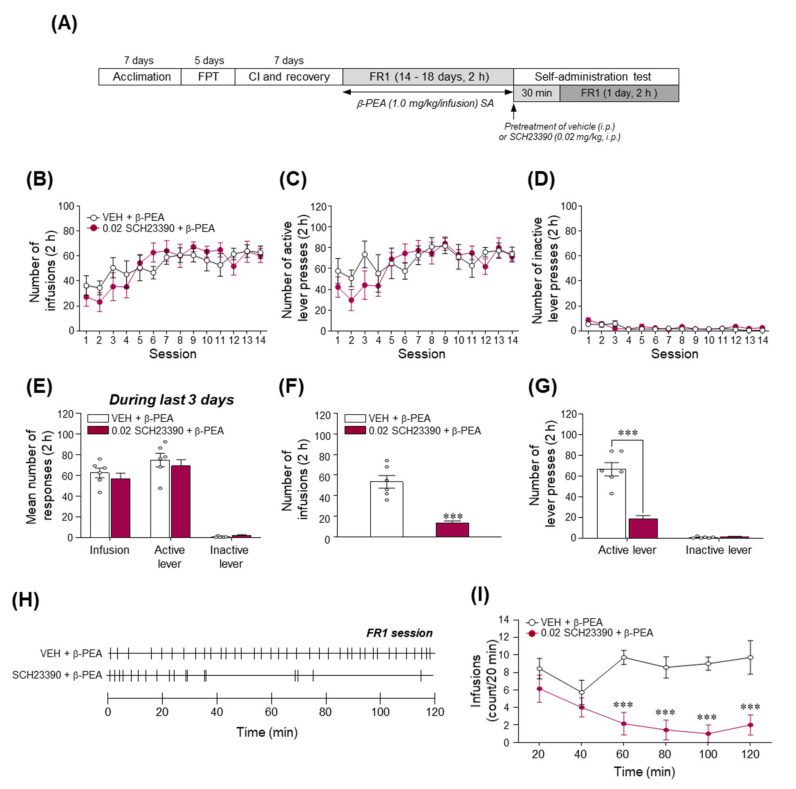
Blockade of the DA D1 receptor significantly attenuated β-PEA-taking behavior in rats. (**A**) Experimental timeline for β-PEA self-administration after SCH23390 (0.02 mg/kg, i.p.) administration. (**B**–**D**) Number of infusions (**B**), active lever pressing (**C**), and inactive lever pressing (**D**) responses during β-PEA self-administration for a 2 h session under FR1 schedules. (**E**–**G**) Number of infusions, active lever pressing, and inactive lever pressing responses during the last three days of the training period of self-administration. (**H,I**) Effect of SCH23390 on the number of infusions (**F**) and active and inactive lever pressing responses (**G**) during β-PEA-taking behavior in self-administration. (**H**) Representative hatch marks indicate the infusion patterns of β-PEA followed by pretreatment with SCH23390. (**I**) Temporal changes in infusions followed by pretreatment with SCH23390. *** *p* < 0.001 vs. saline control group. SAL, saline; β-PEA, β-phenylethylamine. *n* = 6 per group.

## Data Availability

Not applicable.

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
