# Peer review of "Effects of β-Phenylethylamine on Psychomotor, Rewarding, and Reinforcing Behaviors and Affective State: The Role of Dopamine D1 Receptors"

_ijms, 2021, doi:10.3390/ijms22179485_

Round 1

Reviewer 1 Report

The manuscript by Ryu et al. reports on the effects of PEA for a number of behavioral and neurochemical indexes of addictive behavior in rodents.  In general, they find that PEA mimics the effects of other psychomotor stimulants. While a number of these effects have been reported on before, there are some new findings here and the paper is generally well-written and the effects clear.  I have a few minor concerns.

  1. The dose range chosen appears to be too low.  Why was the dose range chosen to include the lowest dose of 3 mg/kg?
  2. I couldn’t find any other studies of rat self-administration. The cited reference (Greenshaw 1984) only refers to an abstract that was never published that I can find.  The authors may want to highlight this new finding.  PEA has been shown to be self-administered in monkeys and dogs that the authors should cite in the introduction.
  3. It is true that the appearance of stereotyped behavior can reduce locomotor activity, but you would typically see increases in locomotor activity at lower doses.
  4. In line 341 insert “ratio” after “progressive”.
  5. Why use mice for some studies and rats for others? It would seem to me rats could have been used for all the studies.
  6. PEA has a short duration of action and Bergmann et al. (Psychopharmacology 2021) showed that inhibiting MAO can facilitate self-administration. This might be worth some discussion.
  7. How many mice were excluded from the CPP study due to initial bias for one side?
  8. In general, how many food sessions were required prior to self-administration training?
  9. How may self-administration sessions were typically required prior to SCH23390 testing?
  10. As pointed out in the introduction, the neurochemical work is correlative. This should also be in the discussion.

Author Response

Response to Reviewer 1

Manuscript number: IJMS-1355784

The authors wish to sincerely thank the reviewers for careful reviews, which, we are certain, have been very useful in further improving the quality of this manuscript. In response to the reviewer’s comments, we have revised the manuscript thoroughly and in accordance with the reviewer’s comments and suggestions. Changes made in the revised manuscript are highlighted in red. We checked and corrected English issues including spells and grammar errors through proofreading of a native speaker throughout the revised manuscript.

Comments

Q1) The dose range chosen appears to be too low. Why was the dose range chosen to include the lowest dose of 3 mg/kg?

Response: Ishida K et al. tested the effect of β-PEA (1.0 ~ 5.0 mg/kg) on the spontaneous discharge of dopamine neurons in the ventral tegmental area1). The results showed that acute systemic administration of β-PEA consistently produced a dose-dependent inhibition of the firing rate of putative dopamine neurons in the ventral tegmental area. In addition, in previous study, 25 mg/kg or 50 mg/kg β-PEA increased extracellular dopamine levels in the nucleus accumbens2) and 50 mg/kg β-PEA increased locomotor activity3). Based on previous studies, we thus tried to test a low dose of 3 mg/kg along with 10, 30 and 50 mg/kg β-PEA on psychomotor and place preference test in present study despite most previous studies used 10-100 mg/kg β-PEA in open-field and conditioned place preference test4-6).

Q2) I couldn’t find any other studies of rat self-administration. The cited reference (Greenshaw 1984) only refers to an abstract that was never published that I can find. The authors may want to highlight this new finding. PEA has been shown to be self-administered in monkeys and dogs that the authors should cite in the introduction.

Response: We have added a reference in the introduction of the revised manuscript (line 50-51 in the revised manuscript).

Q3) It is true that the appearance of stereotyped behavior can reduce locomotor activity, but you would typically see increases in locomotor activity at lower doses.

Response: We agreed with the reviewer’s comment. However, we could not see increased locomotor activity at any doses of β-PEA (n=9/group) in the present study. Thus, we discussed this in Discussion section of the original manuscript.

Q4) In line 341 insert “ratio” after “progressive”.

Response: We have inserted “ratio” after “progressive” as the reviewer’s comment (line 348 in the revised manuscript).

Q5) Why use mice for some studies and rats for others? It would seem to me rats could have been used for all the studies.

Response: It is best that the effects of drugs examine in same species. However, we have limitation on behavior equipment. We have CPP and open-field apparatuses for mice and self-administration apparatus for rats. Thus, we carried out the effects of β-PEA on behaviors, dopamine level and dopamine-related protein changes in rodents (mouse and rats) as our previous studies7,8).

Q6) PEA has a short duration of action and Bergmann et al. (Psychopharmacology 2021) showed that inhibiting MAO can facilitate self-administration. This might be worth some discussion.

Response: We have discussed this in the revised manuscript (line 345-347 in the revised manuscript).

Q7) How many mice were excluded from the CPP study due to initial bias for one side?

Response: We have used 84 mice for pre-conditioning test and excluded 24 mice after pre-conditioning test. Then, we divided into 5 groups (n=12/group). For statistical analysis after post-conditioning test, we excluded the lowest and highest value per each group.  

Q8) In general, how many food sessions were required prior to self-administration training?

Response: We carried out five food sessions. If the rats obtained more than 80 food pellets/1 hr for consecutive three days during five food sessions, the rats were carried out IV catheter implantation surgery.  

Q9) How may self-administration sessions were typically required prior to SCH23390 testing?

Response: As we mentioned in ‘Drug Self-administration’ section of the material and method (line 525-526 in the revised manuscript), rats were self-administered 1.0 mg/kg/infusion β-PEA for at least 14 days prior to SCH23390 test. And, β-PEA self-administration did not over 3 weeks prior to SCH23390 test in the present study.

Q10) As pointed out in the introduction, the neurochemical work is correlative. This should also be in the discussion.

Response: We have added more explanation and references in the revised manuscript as the reviewer’s suggestion (line 302-310 in the revised manuscript).

References

1) Ishida K, Murata M, Katagiri N, Ishikawa M, Abe Kenji, Kato Masatoshi, Utsunomiya I, Taguchi K. Effects of beta-phenylethylamine on dopaminergic neurons of the ventral tegmental area in the rat: a combined electrophysiological and microdialysis study. The Journal of Pharmacology and Experimental Therapeutics 2005;314(2):916-922.

2) Murata M, Katagiri N, Ishida K, Abe K, Ishikawa M, Utsunomiya I, Hoshi K, Miyamoto K, Taguchi K. Effect of β-phenylethylamine on extracellular concentrations of dopamine in the nucleus accumbens and prefrontal cortex. Brain Research 2009;1269:40-46. 

3) Sotnikova TD, Budygin EA, Jones SR, Dykstra LA, Caron MG, Gainetdinov RR. Dopamine transporter-dependent and -independent actions of trace amine beta-phenylethylamine. Journal of Neurochemistry 2004;91(2):362-373.

4) Gilbert D and Cooper SJ. Β-phenylethylamine-, D-amphetamine- and L-amphetamine-induced place preference conditioning in rats. European Journal of Pharmacology 1983;95:311-314.

5) Motahashi N, Nakagawara M, Semba J, Ishii K, Watanabe A, Kariya T. Effect of beta-phenylethylamine on locomotor activity and brain catecholamine metabolism in mice. Yakubutsu Seishin Kodo 1983;3(2):67-75.

6) Dourish CT. A pharmacological analysis of the hyperactivity syndrome induced by beta-phenylethylamine in the mouse. British Journal of Pharmacology 1982;77(1):129-139.

7) Ryu IS, Kim O-H, Lee YE, Kim JS, Li Z-H, Kim TW, Lim R-N, Lee YJ, Cheong JH, Kim HJ, Lee YS, Steffensen SC, Lee BH, Seo J-W, Jang EY. The Abuse Potential of Novel Synthetic Phenylcyclidine Derivative 1-(1-4-Fluorophenyl)Cyclohexyl)Piperidine(4’-F-PCP) in Rodent. International Journal of Molecular Sciences 2020:21(13):4631

8) Ryu IS, Yoon SS, Choi MJ, Lee YE, Kim JS, Kim WH, Cheong JH, Kim HJ, Jang CG, Lee YS, Steffensen SC, Ka M, Woo DH, Jang EY, Seo JW. The potent psychomotor, rewarding and reinforcing properties of 3-fluoromethamphetamine in rodents. Addiction Biology 2020;25(6):e12846

Reviewer 2 Report

This is an excellent manuscript with an impressive amount of data. I only have some minor comments that can be dealt with at the editor’s level.

The title is not representative of the content of the paper. There is a lot more here than experiments on the role of the D1 receptors and there are several other behaviors tested than just addictive behaviors. I suggest a more general title, for example “Effect of ß-phenylethylamine on psychomotor, rewarding and reinforcing behaviors and affective state: role of dopamine D1 receptors”

“western blotting” is usually written with an upper-case W: “Western blotting”

Line 100: delete “exogenous”

Figure 5B: the ticks along the vertical axis do not match up with the numbers

Line 263: the sentence “rats were trained in the active lever pressing associated with drug infusion” is incorrect. I suggest “rat were trained to press the active lever associated with drug infusion”

Line 299: the paper actually has limited evidence of a role of D1 receptors in the dorsal striatum. For example, the authors did not analyze the ventral striatum or PFC, nor did they do micro-injection studies with the D1 antagonist in the dorsal striatum. The conclusion in line 299/300 needs to be more careful, for example “These effects likely involve DAD1R in the dorsal striatum”.

Line 309: correct “with other study” into “with another study”

Line 324/325: circling and head-twitching are not commonly used “schizophrenia-like behaviors”. I suggest to delete “schizophrenia-like”

Line 327: correct “the study” into “a study”

Line 463: it is unclear what is meant with “two boxed minimize exterior noise”.

Line 466: the sentence “Wav files using the microphone connected to computer system and Avisoft-RECORDER software” needs correction, for example “Wav files were analyzed using Avisoft-RECORDER software”

Line 606: correct “staffs” to “staff”

Author Response

Response to Reviewer 2

Manuscript number: IJMS-1355784

The authors wish to sincerely thank the reviewer for careful reviews, which, we are certain, have been very useful in further improving the quality of this manuscript. In response to the reviewer’s comments, we have revised the manuscript thoroughly and in accordance with the reviewer’s comments and suggestions. Changes made in the revised manuscript are highlighted in red. We checked and corrected English issues including spells and grammar errors through proofreading of a native speaker throughout the revised manuscript.

Comments

This is an excellent manuscript with an impressive amount of data. I only have some minor comments that can be dealt with at the editor’s level.

Q1) The title is not representative of the content of the paper. There is a lot more here than experiments on the role of the D1 receptors and there are several other behaviors tested than just addictive behaviors. I suggest a more general title, for example “Effect of ß-phenylethylamine on psychomotor, rewarding and reinforcing behaviors and affective state: role of dopamine D1 receptors”

Response: As the reviewer suggested, we have changed the title.

Q2) “western blotting” is usually written with an upper-case W: “Western blotting”

Response: We have changed all ‘western’ to ‘Western’ in the revised manuscript. 

Q3) Line 100: delete “exogenous

 Response: We have deleted ‘exogenous’ in the revised manuscript.

Q4) Figure 5B: the ticks along the vertical axis do not match up with the numbers

 Response: We have fixed this point and we have changed ‘Figure 5B’ in the revised manuscript.

Q5) Line 263: the sentence “rats were trained in the active lever pressing associated with drug infusion” is incorrect. I suggest “rat were trained to press the active lever associated with drug infusion”

Response: It is correct. We have changed the sentence as the reviewer’s comment (line 262-263 in the revised manuscript).

Q6) Line 299: the paper actually has limited evidence of a role of D1 receptors in the dorsal striatum. For example, the authors did not analyze the ventral striatum or PFC, nor did they do micro-injection studies with the D1 antagonist in the dorsal striatum. The conclusion in line 299/300 needs to be more careful, for example “These effects likely involve DAD1R in the dorsal striatum”.

Response: We agree with the reviewer’s comment. We have changed the conclusion as the reviewer’s comment (line 297 in the revised manuscript).

Q7) Line 309: correct “with other study” into “with another study”

Response: We have changed “other” into “another” as the reviewer’s comment (line 313 in the revised manuscript).

Q8) Line 324/325: circling and head-twitching are not commonly used “schizophrenia-like behaviors”. I suggest to delete “schizophrenia-like”

Response: As reviewer’s comment, we have deleted ‘schizophrenia-like’ in the revised manuscript.

Q9) Line 327: correct “the study” into “a study”

Response: We have changed “the future study” into “a future study” as the reviewer’s comment (line 330 in the revised manuscript).

Q10) Line 463: it is unclear what is meant with “two boxed minimize exterior noise”.

Response: We have fixed this sentence in the revised manuscript: two boxes to minimize exterior noise (line 470 in the revised manuscript).  

Q11) Line 466: the sentence “Wav files using the microphone connected to computer system and Avisoft-RECORDER software” needs correction, for example “Wav files were analyzed using Avisoft-RECORDER software”

Response: We have changed the sentence in the revised manuscript (line 473 - 474 in the revised manuscript).    

Q12) Line 606: correct “staffs” to “staff”

Response: We have deleted ‘s’ in the ‘staffs’ (line 612 in the revised manuscript).